# Climate-Related Heat Stress and Psychological Outcomes in Self-Employed Delivery Workers: Evidence from Brasília, Brazil

**DOI:** 10.3390/ijerph22111666

**Published:** 2025-11-03

**Authors:** Carlos Manoel Lopes Rodrigues, Lígia Abreu Gomes Cruz

**Affiliations:** 1Postgraduate Program in Psychology, University Center of Brasília (CEUB), Brasília 70713-000, Brazil; 2Postgraduate Program in Clinical Psychology and Culture (PPG PsiCC), University of Brasília (UnB), Brasília 70910-900, Brazil; 3Institute of Psychology, Federal University of Uberlândia (UFU), Uberlândia 38400-902, Brazil; ligia.abreu@ufu.br

**Keywords:** ecological momentary assessment, occupational heat, delivery workers, psychosocial risk

## Abstract

This study examines whether daily heat exposure worsens psychological well-being among self-employed motorcycle delivery workers in Brasília, Brazil. Using ecological momentary assessment over 15 consecutive days in August 2025, 45 workers were recruited and 30 (66.7%) completed twice-daily mobile prompts (12:00 and 18:00) rating stress, fatigue, mood, and perceived heat (1–5 scales) and reporting kilometers traveled. Environmental data (temperature, relative humidity, barometric pressure) were paired from the INMET Brasília station. Linear regressions with cluster-robust standard errors by participant tested associations. Higher temperature was consistently related to greater strain: each +1 °C was associated with higher stress (β = 0.196, 95% CI 0.179–0.213), higher fatigue (β = 0.289, 95% CI 0.284–0.295), and worse mood (β = 0.149, 95% CI 0.130–0.168). Adding relative humidity yielded small but reliable partial effects (lower stress and better mood, yet higher fatigue) amid strong dry-season collinearity between temperature and humidity. The findings indicate that even modest day-to-day warming corresponds to measurable deterioration in psychological outcomes in a precarious, outdoor, platform-mediated workforce. Policies that expand hydration and shaded rest access, integrate heat indices into alerts, and adapt platform scheduling to reduce peak-heat exposure may mitigate risk.

## 1. Introduction

Climate change has emerged as one of the most pressing global challenges of the 21st century, with rising ambient temperatures and extreme heat events posing direct threats to human health, safety, and productivity [1,2,3]. A growing body of evidence shows that heat exposure is not only a physiological hazard but also a psychosocial and mental health risk factor, affecting stress, mood, cognitive performance, and accident rates [4,5,6]. While the literature has traditionally emphasized the physical consequences of heat, recent reviews underscore the relevance of psychological and behavioral dimensions of occupational heat strain [7,8].

In Brazil, the challenges are compounded by rapid urban warming, demographic inequalities, and a large share of outdoor and precarious work [9,10]. Informal and self-employed delivery workers are particularly vulnerable, as they often operate under extreme heat without institutional protection, stable labor contracts, or adequate occupational safety measures [11,12]. These workers have become emblematic of the platform economy, marked by intense labor demands, low wages, and high turnover. Studies show that delivery riders and drivers typically work long shifts that frequently exceed 49–60 h per week, with a significant portion reporting more than 60 h, conditions far above the national average for self-employed workers [13,14]. Despite such workloads, their average monthly income remains well below national standards, hovering around R$1600 for app-based couriers—more than R$1000 below the national average. Moreover, less than one quarter of these workers contribute to social security, compared to nearly half of non-platform delivery workers, reflecting both structural vulnerability and the erosion of formal employment in the sector.

The rise of platform-mediated labor has thus been accompanied by a broader process of precarization, in which algorithmic management controls task allocation, pricing, and even working hours, while sustaining a discourse of autonomy and entrepreneurship. In practice, the majority of app-based drivers and riders report little control over pay, clients, or schedules, as their activities are heavily influenced by digital platforms through penalties, incentives, and suggested shifts [13,15,16]. This contradiction has been described as an “illusion of autonomy,” masking the subordination and heightened risks faced by these workers [17].

Heat is traditionally classified as a physical hazard within occupational health, yet it can also acquire a psychosocial dimension when combined with precarious working arrangements such as those experienced by delivery workers. In this field, psychosocial risk factors are defined as the organizational and social conditions of work that affect health, well-being, and performance, including aspects such as excessive workloads, low autonomy, and job insecurity [18]. The interaction between environmental exposure and structural features of platform-mediated labor, including long shifts, income insecurity, algorithmic control, and limited recovery opportunities, transforms heat into a dual risk factor: both a physiological stressor and a psychosocial stressor. Importantly, this transformation does not occur in a vacuum but is shaped by deep social inequalities [2,10,19].

Although heat exposure has been consistently linked to declines in productivity and increases in accidents, absenteeism, and morbidity [20,21,22,23,24,25], empirical evidence on its psychological impacts in informal urban labor markets remains limited. Controversies remain about the magnitude of mental health effects, with some studies reporting strong associations between temperature and mental distress [6,26], while others emphasize socioeconomic vulnerabilities as the primary determinants of risk [27,28]. Moreover, most of the current research has focused on formal occupational groups such as construction or agriculture [29,30], leaving urban gig workers understudied despite their critical exposure profile.

The present pilot study addresses this gap by examining the effects of climate-related heat exposure on psychological outcomes among self-employed delivery workers in Brasília, Brazil. This group represents a relevant case for understanding how informal labor, precarious conditions, and climate stressors interact to shape occupational health risks. By combining daily environmental measurements with repeated self-reports of stress, fatigue, and mood, the study shows that even modest temperature increases are associated with measurable declines in psychological well-being.

This research adds to existing literature by focusing on a worker group that has received limited attention in studies of heat and mental health. Most previous investigations have concentrated on formal sectors such as construction and agriculture, where exposure conditions and employment relationships are clearly defined and regulated. In contrast, the present study examines self-employed delivery workers in an urban setting, whose activities occur under informal and often unprotected labor arrangements, in which environmental and psychosocial risks coexist.

## 2. Materials and Methods

This pilot study adopted a repeated-measures observational design using ecological momentary assessment (EMA) to examine the effects of climate-related heat exposure on psychological outcomes among self-employed delivery workers in Brasília, Brazil. EMA is a method that samples participants’ experiences in real time and in their natural environments, minimizing recall bias and capturing within-person fluctuations across the day. We operationalized EMA by sending automated mobile prompts twice daily—at 12:00 and 18:00 local time—over 15 consecutive days in August 2025 (Brasília’s dry season).

A total of 45 motorcycle couriers were initially recruited through community networks of couriers operating with food and parcel delivery platforms; a total of 30 (66.7%) completed the full protocol and were included in the analyses. Of the 45 delivery workers initially recruited, 30 (66.7%) completed at least 70% of the ecological momentary assessment prompts and were retained for analysis. Five participants withdrew during the first week, and ten were excluded due to insufficient data. A comparison of completers and non-completers on age and work experience indicated no significant differences (all *p* > 0.10), suggesting that attrition was not systematically related to these variables.

The final sample (*n* = 30) provided 900 repeated observations (30 participants × 15 days × 2 prompts per day), ensuring adequate power (1 − β = 0.80, α = 0.05) to detect small-to-moderate within-person effects (f^2^ ≈ 0.02) according to EMA design standards.

Participants reported an average body height of 1.74 m (SD = 0.07), weight of 75.9 kg (SD = 9.8), and body mass index (BMI) of 25.1 kg/m^2^ (SD = 2.4). Most participants had completed high school (73.3%), while 26.7% had some tertiary education. Work experience in delivery services ranged from 1 to 12 years (M = 5.4, SD = 3.1), and the average weekly workload was approximately 58.7 h (SD = 9.4). Based on the Compendium of Physical Activities [31,32], the typical tasks performed (motorcycle riding, loading and unloading items, and short walking intervals) correspond to an estimated energy expenditure of 4.0–5.5 metabolic equivalents (METs), or roughly 230–330 W for a 70 kg individual, indicating a moderate-intensity metabolic rate sustained over extended daily exposure periods.

Psychological measures (EMA scales). At each prompt, participants rated four psychological states—stress, fatigue, mood, and perceived heat strain—using single-item EMA scales on a five-point Likert-type response format (1 = not at all to 5 = extremely). The items were phrased as follows: “How stressed do you feel right now?”, “How tired do you feel right now?”, “How is your mood right now?”, and “How hot do you feel right now while working?”. The mood item was reverse-coded so that higher values represented poorer mood. These items were adapted from validated EMA procedures in occupational and environmental psychology [32,33], chosen for their brevity and ecological validity in capturing momentary psychological fluctuations in real-world settings.

The use of brief items reduces participant burden while effectively capturing real-time fluctuations in psychological states. All items were pilot-tested with a subset of delivery workers (*n* = 6) to ensure comprehension and contextual relevance. Given the high mobility of motorcycle delivery workers, the use of short, context-specific single-item measures minimized participant burden and maximized compliance without compromising construct validity. All items were pilot-tested with a small group of riders (*n* = 6) to ensure clarity and relevance to the occupational context. Reliability was examined through multilevel variance components estimated from random-intercept models, which revealed ICC(1) and ICC(2) values below 0.01 for all scales—indicating that nearly all variance occurred within individuals across time, as expected for momentary state assessments.

We opted not to use composite heat stress indices such as the Wet Bulb Globe Temperature (WBGT), the Universal Thermal Climate Index (UTCI), or the Thermal Sensation Vote (TSV) because our objective was not to estimate biophysical strain under controlled conditions, but rather to capture how workers themselves experience heat in real-world contexts. These indices, while valuable in occupational hygiene and ergonomics, often require instrumentation, microclimatic monitoring, or metabolic assumptions that are difficult to implement in informal and highly mobile work settings such as motorcycle delivery.

Moreover, indices like WBGT or the Heat Stress Index (HSI) provide standardized estimates of physiological load but are impractical for self-employed, constantly moving riders who lack fixed worksites or protective monitoring infrastructure. Our goal was to examine subjective heat perception as a psychosocial risk factor, rather than physiological heat strain per se. To ensure comprehension and reliability, the subjective scales used were adapted from validated Likert-type measures previously employed in occupational and environmental psychology research, and participants received brief, standardized instructions with examples during recruitment. Thus, the perceived heat strain scale reflects participants’ immediate, ecologically valid experience of thermal discomfort during actual work shifts, complementing rather than replacing objective measures used in other occupational settings.

EMA, by contrast, allowed us to combine meteorological station data with workers’ subjective perceptions and psychological states, generating a context-sensitive dataset that reflects both environmental exposure and lived experience. This approach aligns with the study’s focus on psychosocial risk, prioritizing the intersection between physical conditions and psychosocial strain in precarious labor markets.

Environmental variables were obtained from the Instituto Nacional de Meteorologia (INMET) through the official Brasília meteorological station (https://portal.inmet.gov.br/, accessed on 17 August 2025). For each day and time point, corresponding values of temperature (°C), relative humidity (%), and barometric pressure (hPa) were extracted from the INMET dataset and paired with the participants’ self-reports. This ecological momentary assessment design yielded 900 observations (30 workers × 15 days × 2 prompts), providing repeated within-person measures across varying thermal conditions.

Associations between temperature and psychological outcomes were tested using linear regression models estimated separately for each dependent variable, with cluster-robust standard errors by participant to account for the non-independence of repeated measures. This approach was chosen because the data followed an ecological momentary assessment (EMA) structure with unbalanced repeated observations across participants, for which repeated-measures ANOVA and post hoc tests would be inappropriate due to their assumptions of balanced cells and categorical predictors.

Predictors were treated as continuous variables, and model robustness to heteroskedasticity and within-person correlation was ensured through heteroskedasticity- and cluster-robust (sandwich) standard errors. The assumptions of normality and homogeneity of variance were inspected visually through Q–Q and scale–location plots; no substantial violations were observed. Because standard homogeneity tests such as Levene’s are not applicable in this continuous-predictor, repeated-measures context, robust estimators were preferred.

Bivariate associations among environmental and psychological variables were examined using Spearman’s rank correlation coefficients (ρ), given the ordinal (5-point) scaling of the outcomes. The same analytical pattern and direction of effects were obtained when Pearson’s r was tested, confirming robustness. Humidity and barometric pressure were evaluated as covariates but did not materially change the estimates and were omitted from the final models for parsimony. Results are reported as unstandardized coefficients with 95% confidence intervals and two-tailed *p*-values. All analyses were conducted in Rversion 4.5.1; R Foundation for Statistical Computing, Vienna, Austria).

The study protocol was approved by the Research Ethics Committee of Centro Universitário de Brasília (Approval Code: 89798625.0.0000.0023) and adhered to the Declaration of Helsinki. The dataset and statistical code (Appendix A) are available as Appendix A at the Open Science Framework (OSF): https://doi.org/10.17605/OSF.IO/Y6HDC (accessed on 12 September 2025).

## 3. Results

### 3.1. Environmental Conditions

During the 15-day observation period, daily air temperature and relative humidity exhibited marked diurnal variability consistent with Brasília’s dry-season pattern. Temperatures ranged from 16.8 °C to 28.5 °C, while relative humidity fluctuated between 19% and 71% (Table 1).

As shown in Figure 1, temperature values were consistently higher at 12:00 compared to 18:00, reflecting intense solar radiation and low wind activity during midday hours. Relative humidity followed an inverse pattern, reaching its lowest levels at 12:00 when temperatures peaked, and recovering slightly by 18:00. This pattern reflects the characteristic continental dryness and high solar radiation of Brasília’s winter, when sky conditions are typically cloudless. Meteorological records from the same period indicated no rainfall and a mean daily insolation of approximately 9 h, representing intense solar exposure throughout working hours.

### 3.2. Stress

Daily self-reports of stress showed moderate variability (M = 3.85, SD = 1.24), with scores ranging from 2.67 to 4.97 throughout the 15-day observation period. At 12:00, mean stress levels were lower (M = 3.12, SD = 0.47), while at 18:00 they rose consistently (M = 4.62, SD = 0.31), reflecting an accumulation of physical and psychological strain over the course of the workday.

The lowest stress levels were recorded on cooler days, such as August 3 (16.8 °C, M = 2.97) and August 10 (18.1 °C, M = 2.80), while the highest levels occurred during the hottest afternoons, including August 6 (26 °C, M = 4.97) and August 8 (27.8 °C, M = 4.97). Across the observation window, the pattern of daily means suggested a close alignment between heat exposure and elevated stress responses, with stress peaking on days when air temperature exceeded 27 °C and declining when conditions were cooler (Figure 2).

### 3.3. Fatigue

Fatigue presented a similar pattern (M = 3.81, SD = 1.29), with values ranging from 1.37 to 5.00 across the 15-day observation period. Morning reports at 12:00 indicated lower fatigue (M = 2.73, SD = 0.97), while afternoon reports at 18:00 consistently reached the upper end of the scale (M = 4.89, SD = 0.19), showing a marked increase throughout the day.

The lowest fatigue scores were recorded on cooler days such as August 3 (16.8 °C, M = 1.67) and August 12 (17.8 °C, M = 1.37), while maximum fatigue values occurred on the hottest afternoons, including August 6–9, when temperatures ranged between 26.0 °C and 28.2 °C and mean fatigue scores reached 5.00. This pattern indicates that riders frequently experienced intense fatigue after midday deliveries, coinciding with the period of highest thermal load (Figure 3).

### 3.4. Mood

Mood ratings averaged 3.04 (SD = 1.40), with scores ranging from 2.00 to 4.67 across the 15-day period, where higher values indicate poorer mood. At 12:00, mean mood scores were generally lower (M = 2.42, SD = 0.33), while at 18:00 they increased (M = 3.72, SD = 0.49), suggesting a gradual deterioration in affective state as the workday progressed.

The most positive moods were reported on cooler, more humid days such as August 3 (16.8 °C, 53% RH, M = 2.70) and August 11 (18.7 °C, 41% RH, M = 2.00), whereas the most negative moods were observed during hotter and drier conditions, particularly between August 6 and 9, when afternoon temperatures reached 27–28 °C and mood ratings approached 4.5–4.7 (Figure 4).

### 3.5. Correlational Analysis

Correlation analyses indicated significant relationships between environmental and psychological variables across the 15-day observation period (Figure 5). Higher temperature was positively correlated with stress (ρ = 0.60, 95% CI [0.42, 0.74], *p* < 0.001), fatigue (ρ = 0.90, 95% CI [0.83, 0.94], *p* < 0.001), mood deterioration (ρ = 0.38, 95% CI [0.12, 0.59], *p* < 0.001), and perceived heat strain (ρ = 0.93, 95% CI [0.88, 0.96], *p* < 0.001).

Conversely, relative humidity showed negative correlations with all psychological outcomes, including stress (ρ = −0.50, 95% CI [−0.67, −0.28], *p* < 0.001), fatigue (ρ = −0.64, 95% CI [−0.77, −0.46], *p* < 0.001), mood (ρ = −0.34, 95% CI [−0.56, −0.07], *p* < 0.001), and perceived heat strain (ρ = −0.72, 95% CI [−0.82, −0.56], *p* < 0.001). Finally, temperature and relative humidity were inversely correlated (ρ = −0.70, 95% CI [−0.81, −0.53], *p* < 0.001.

### 3.6. Regression Analysis

Regression models with temperature as the sole predictor confirmed significant linear associations with all psychological outcomes. Each 1 °C increase in air temperature was associated with higher stress (β = 0.196, 95% CI [0.179, 0.213], *p* < 0.001), greater fatigue (β = 0.289, 95% CI [0.284, 0.295], *p* < 0.001), and poorer mood (β = 0.149, 95% CI [0.130, 0.168], *p* < 0.001). Model fit was highest for fatigue (R^2^ = 0.800), followed by stress (R^2^ = 0.395) and mood (R^2^ = 0.181).

When relative humidity was added to the model, the overall fit improved slightly, and partial effects diverged from the bivariate correlations due to the negative collinearity between temperature and humidity (Table 2). Controlling for temperature, higher humidity was associated with lower stress (β = –0.010, 95% CI [–0.017, –0.004], *p* = 0.002) and better mood (β = –0.011, 95% CI [–0.019, –0.003], *p* = 0.007), but with greater fatigue (β = 0.016, 95% CI [0.013, 0.020], *p* < 0.001).

Model diagnostics indicated that residuals were normally distributed (Shapiro–Wilk *p* = 0.273) and variances were homogeneous (Levene’s *p* = 0.338). Cluster-robust standard errors were applied to account for repeated measures within participants. No evidence of problematic multicollinearity was found beyond the expected temperature–humidity overlap (Variance Inflation Factor < 2.0).

Figure 6 displays the fitted regression models describing the effects of air temperature and relative humidity on psychological outcomes. Each 1 °C increase in temperature was associated with an average rise of 0.17 units in stress (*p* < 0.001), 0.33 units in fatigue (*p* < 0.001), and 0.12 units in mood deterioration (*p* < 0.001). In contrast, higher relative humidity predicted lower stress (−0.01 units, *p* = 0.002) and better mood (−0.01 units, *p* = 0.007), but slightly higher fatigue (+0.02 units, *p* < 0.001). Among the three outcomes, fatigue showed the steepest slope, followed by stress and mood, indicating greater susceptibility of physical exhaustion to ambient heat

## 4. Discussion

Although the mean ambient temperature recorded at the INMET station was 23.25 °C, this value likely underestimates the real thermal stress experienced by delivery workers in Brasília. The city, located on the Central Plateau (15° S), presents intense solar radiation—often exceeding 950 W/m^2^ at midday—and extremely low humidity (frequently below 20% during the dry season). Riders operate in direct sunlight on asphalt surfaces that may reach 50–60 °C, with additional radiant heat from engines and limited airflow in traffic. They commonly wear long trousers, closed shoes, and synthetic jackets for protection, covering 70–100 km daily. These combined factors—solar radiation, clothing insulation, engine proximity, and sustained exertion—create microclimatic heat loads far higher than those indicated by ambient measures, supporting the plausibility of measurable heat strain despite moderate mean air temperatures.

Our findings are consistent with reviews demonstrating that occupational heat exposure compromises health, cognition, and productivity [5,7,8] and extend meta-analytic evidence linking ambient heat to mental health outcomes [2,6]. The strong temperature–fatigue association aligns with thermoregulatory and cardiovascular strain mechanisms that elevate perceived exertion and deplete energy resources during heat exposure [5]. Likewise, the positive associations between temperature, stress, and mood deterioration correspond to previous findings on irritability, sleep disturbance, and affective dysregulation under heat [28,34] and echo the economic and safety consequences documented in high-temperature workplaces [4,24].

The mixed effects of humidity—protective for stress and mood but adverse for fatigue—likely reflect two mechanisms: the strong inverse temperature–humidity relationship typical of Brasília’s dry season and distinct biophysical processes. Higher humidity reduces evaporative cooling, intensifying physiological fatigue, while slightly cooler and more humid periods may coincide with less psychologically aversive conditions. This pattern is consistent with ergonomics models such as WBGT and UTCI, in which similar temperatures can impose different total heat loads depending on moisture levels [26,29].

In Brazil, urban warming and social inequalities magnify heat-related vulnerability [9,10]. Delivery workers experience prolonged outdoor exposure, traffic hazards, and minimal institutional protection. The phantomization of labor has intensified working hours, income instability, and social insecurity [13,15]. Algorithmic management of schedules and pay may further constrain recovery opportunities and encourage risk-taking during heat exposure [14,16]. These structural pressures plausibly amplify the psychological burden of heat observed in this study and help explain the pronounced fatigue effects.

The relationship between heat and road safety is also critical. Heat exposure has been associated with increased crash risk and severity [22,25], including among motorcyclists [23]. In a workforce traveling long distances daily, heat-related fatigue, irritability, and attentional decline may elevate accident risk, introducing a public-safety dimension to what is often treated merely as a comfort or productivity issue.

By documenting within-person psychological deterioration among informal, urban delivery workers, this study extends evidence on occupational heat stress—traditionally centered on agriculture and construction [20,29,30]. While structural disadvantage contributes to vulnerability [27,30], our design demonstrates that day-to-day temperature variation independently explains meaningful within-worker changes in stress, fatigue, and mood. This underscores both environmental and labor-related levers for intervention in contemporary urban economies.

Our results argue for multi-layered adaptation spanning urban planning, public health, and platform design:Workplace and public-health measures: routine hydration access, shaded rest points, cooled refuges, and heat alert systems targeting outdoor workers [5,7].Scheduling and algorithmic nudges: platform-level mitigation (e.g., dynamic de-intensification of order dispatch during peak heat; incentives for breaks; minimum recovery intervals; route optimization to reduce exposure).Standards and indices: adopt heat indices appropriate for tropical cities (e.g., WBGT/UTCI) in regulations and alerts, rather than temperature alone [26,29].Social protection: expand coverage (e.g., social security, sickness protection) to reduce the economic compulsion to work through hazardous heat [13,16].Safety integration: couple heat alerts with road-safety messaging for riders and drivers, given evidence linking heat to crash risk [23,25].

Limitations should temper inference. First, weather was measured at the station level; microclimates along routes (e.g., urban heat islands, radiant load, wind) were not directly captured, potentially biasing point estimates toward the null or introducing exposure misclassification. Second, outcomes were self-reported; although EMA reduces recall bias, common-method variance and expectancy effects remain possible. Third, selection and completion: 45 were recruited and 30 completed the protocol; differential attrition could bias estimates if non-completers differed systematically. Fourth, time-of-day and workload composition (e.g., mid-day vs. evening, traffic cycles) might confound associations; although cluster-robust SEs address non-independence, future models should incorporate time-fixed effects, distance traveled, and shift-level characteristics. Fifth, strong temperature–humidity collinearity in the dry season complicates estimation of independent humidity effects; non-linear and interaction terms (e.g., temperature × humidity) and heat-stress indices could yield more interpretable parameters. Finally, causality cannot be established in an observational design; however, the magnitude and consistency of within-worker associations are noteworthy and align with mechanistic plausibility.

Future research, priority directions include the following:High-resolution exposure: wearable sensors for personal temperature, humidity, radiant load, sweat rate, and heart rate; integration with route-level microclimate mapping.Advanced modeling: multilevel mixed models, generalized additive models for nonlinearity/thresholds, and distributed-lag structures to quantify delayed effects.Psychophysics tests: directly compare linear, logarithmic (Fechner), and power-law (Stevens) mappings for perceived heat vs. temperature, and assess mediation of stress, mood, and fatigue by perceived heat.Indices and interventions: evaluate WBGT/UTCI triggers for automated platform nudges, randomized hydration/shade/rest protocols, and algorithmic throttling during heatwaves.External outcomes: link EMA to crash/near-miss reports and platform telematics to quantify safety implications under heat.Comparative contexts: replicate across seasons, cities, and platform types (motorcycle, bicycle, e-bike) and examine heterogeneity by socioeconomic status and social protection.

## 5. Conclusions

This study demonstrates that even modest increases in daily temperature are associated with measurable declines in psychological well-being among delivery workers operating under informal, platform-mediated labor conditions. By analyzing within-person variations using ecological momentary assessment, the study extends heat–health research beyond formal sectors such as construction and agriculture. Protecting these workers will require coordinated efforts in public health, regulation, and platform design to reduce exposure, support recovery, and strengthen social protection.

## Figures and Tables

**Figure 1 ijerph-22-01666-f001:**
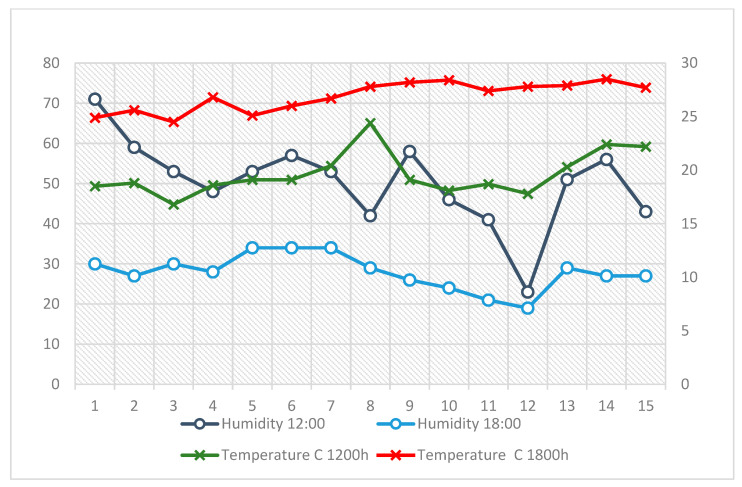
Daily fluctuations in air temperature (°C) and relative humidity (%RH) recorded at 12:00 and 18:00 across the 15-day observation period in Brasília’s dry season.

**Figure 2 ijerph-22-01666-f002:**
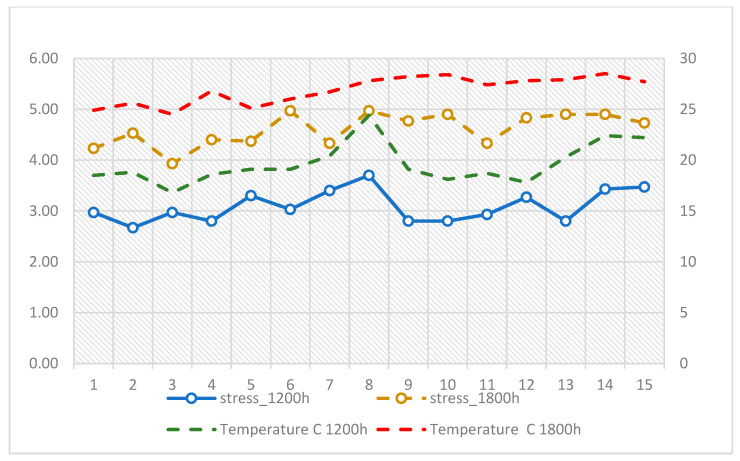
Daily variation in stress at 12:00 and 18:00 in relation to air temperature (°C) across the 15-day observation period.

**Figure 3 ijerph-22-01666-f003:**
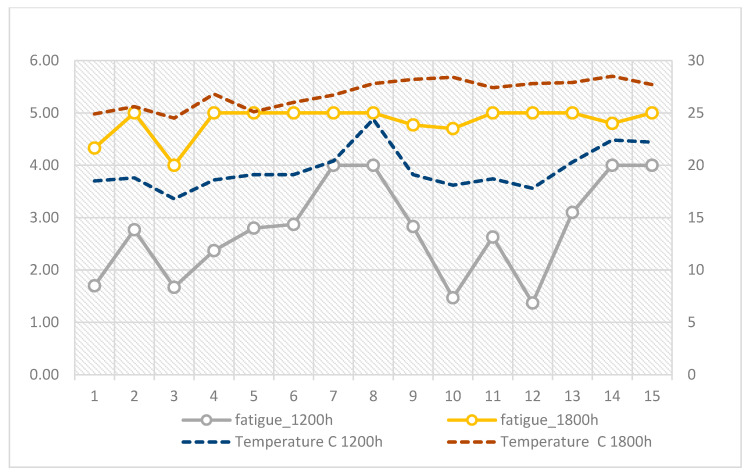
Daily variation in fatigue at 12:00 and 18:00 in relation to air temperature (°C) across the 15-day observation period.

**Figure 4 ijerph-22-01666-f004:**
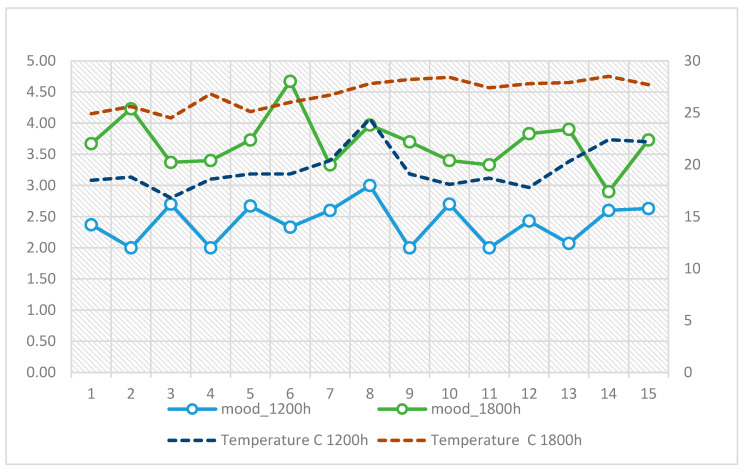
Daily variation in mood at 12:00 and 18:00 in relation to air temperature (°C) across the 15-day observation period.

**Figure 5 ijerph-22-01666-f005:**
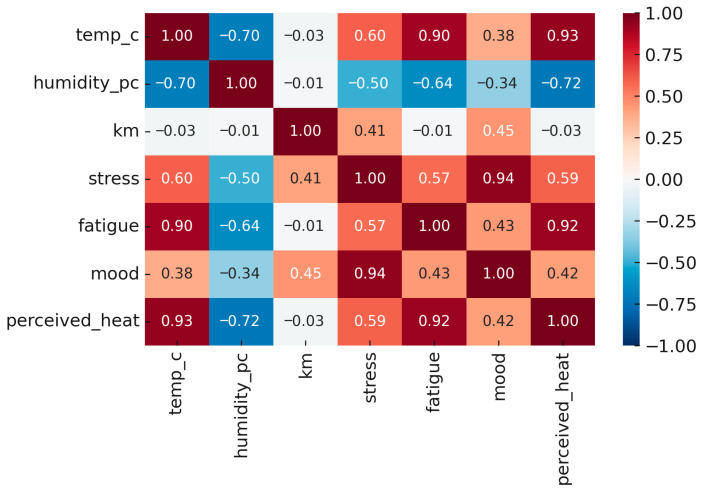
Correlation matrix of environmental and psychological variables.

**Figure 6 ijerph-22-01666-f006:**
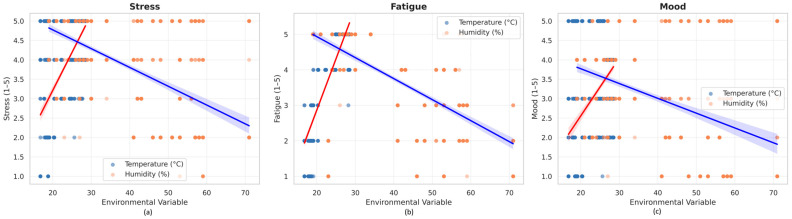
Regression models of psychological outcomes as a function of air temperature (°C) and relative humidity (%). The schemes follow a three-panel format: (**a**) Stress, (**b**) Fatigue, and (**c**) Mood. Shaded areas represent 95% confidence intervals. Red lines indicate fitted regression curves for air temperature, and blue lines indicate fitted curves for relative humidity.

**Table 1 ijerph-22-01666-t001:** Descriptive statistics.

Variable	M	SD	Min	Max
Temperature	23.25	3.99	16.80	28.50
Relative humidity	39.10	13.73	19.00	71.00
Distance	72.96	16.41	45.00	100.00

M = Mean; SD = Standard Deviation. Temperature in degrees Celsius (°C). Distance in kilometers (km). Psychological outcomes were rated on 1–5 scales, with higher scores indicating greater stress, fatigue, mood deterioration, or perceived heat.

**Table 2 ijerph-22-01666-t002:** Linear regression models predict psychological outcomes from temperature (°C) and relative humidity (%).

Outcome	Temperature			Humidity %			R^2^
	β	95% CI	*p*	β	95% CI	*p*	
Stress	0.168	[0.144, 0.193]	0.000	−0.010	[−0.017, −0.004]	0.002	0.400
Fatigue	0.334	[0.322, 0.345]	0.000	0.016	[0.013, 0.020]	0.000	0.812
Mood	0.119	[0.090, 0.147]	0.000	−0.011	[−0.019, −0.003]	0.007	0.186

Note. Models estimated with cluster-robust standard errors by participant. Reported are unstandardized coefficients (β), 95% confidence intervals (CI), *p*-values, and R^2^.

## Data Availability

The dataset and statistical code are openly available at the Open Science Framework (OSF): https://doi.org/10.17605/OSF.IO/Y6HDC (accessed on 12 September 2025).

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
