# Peer review of "Climate-Related Heat Stress and Psychological Outcomes in Self-Employed Delivery Workers: Evidence from Brasília, Brazil"

_ijerph, 2025, doi:10.3390/ijerph22111666_

Round 1

Reviewer 1 Report

Comments and Suggestions for Authors

The title and the abstract of this work look interesting. However, I was somewhat disappointed when I looked into this manuscript. My detailed comments are given as below.

  1. The major concern is that the mean air temperature was 23.25 degree C and the mean RH was 39%. Are you sure such conditions would trigger a heat strain??? The metabolic rates of these participants were missing in this study. I therefore could not judge the activity levels of participants. Consequently, the ‘heat stress’ reported in this study may not exist or not reliable, since either the environmental temperature or the activity levels were not sufficient to trigger a heat strain.
  2. Page 3. Brief information of these participants should be provided, including their mean age, height, BMI, working experiences, as well as some other factors like educational backgrounds, their daily metabolic rates.
  3. Please remove PMV here since it is a thermal sensation measure for indoor steady environment. It is not a heat stress index.
  4. Please clarify why these common heat stress indices (like WBGT, HSI) were not used in this study. These indices are more reliable than results from the perceived heat strain scale. The stress levels acquired in this study were from self-reported results of participants, rather than physiological indicators like core temperature and HSI. How could you be sure all participants could understand these subjective scales well?

  1. In the methodology section, details of these mental scales for the measurements of stress, mood, and perceived heat strain were missing. Please provide the details of these scales, please also justify them.
  2. The complete statistical analysis methods should be provided in the methodology section, as well as the complete procedures, rather than simply stating that the LRM was used without a single justification. In addition, some important procedures, like the normality test, and the homogeneity of variances test were missing. Was repeated measures ANOVA used in this study? What's the pot-hoc method used in this study? What correlation analysis method did you used? Judging from the correlation coefficients r in result section, I guess you used the Pearson’s correlation analysis method. However, the variables in your study were discrete ones, which meant that it is not proper to use Pearson’s correlation method.
  3. For the result section, the results were not sufficiently reported. Please provide the air temperature and RH data during these 15 days firstly. Then please present detailed changes of the self-reported stress, fatigue, and the mood one by one. It is strongly recommenced to detail these 3 outcomes in 3 sections with subtitles. Then the correlation analysis result followed. I also suggest providing the regression results, including the curves and specific outcomes, instead of just providing a Table 2.
  4. The first paragraph in Page 4 was not necessary. There was no subheadings in this section.
  5. The last paragraph of the result section seemed the presentation of the coupling effects of the heat and humidity. Please expand this paragraph, by including more detailed and specific results. Please also insert figures and provide statistical results.
  6. In the first sentence of the discussion, the ambient temperature in this study was not high at all.

Author Response

Comments 1: The major concern is that the mean air temperature was 23.25 degree C and the mean RH was 39%. Are you sure such conditions would trigger a heat strain??? The metabolic rates of these participants were missing in this study. I therefore could not judge the activity levels of participants. Consequently, the ‘heat stress’ reported in this study may not exist or not reliable, since either the environmental temperature or the activity levels were not sufficient to trigger a heat strain.

Response 1: We thank the reviewer for this valuable observation. Although the mean ambient temperature registered at the INMET station (23.25°C) may appear mild in isolation, this value underestimates the effective heat load experienced by delivery workers in Brasília’s microclimatic context. Brasília has one of the highest solar radiation levels in Brazil, frequently exceeding 950 W/m² at midday, combined with very low relative humidity (often below 20% during the dry season). The riders’ thermal exposure is intensified by direct solar incidence on asphalt surfaces that can reach 50–60°C, radiant heat from motorcycle engines, and the use of protective gear (jeans, jackets, and helmets) that restrict convective and evaporative heat dissipation. These workers also sustain high physical and metabolic demands, traveling an average of 70–100 km per day, often under time pressure and with limited recovery opportunities.

Therefore, while the ambient temperature measured at the meteorological station represents a regional mean, the microclimatic and metabolic conditions typical of motorcycle couriers in Brasília plausibly generate significant heat strain. We have clarified this context in the manuscript (Section 3, Results) and inserted a methodological note emphasizing that environmental heat indices do not capture the combined thermal and metabolic load of this occupational setting.

Comments 2: Page 3. Brief information of these participants should be provided, including their mean age, height, BMI, working experiences, as well as some other factors like educational backgrounds, their daily metabolic rates.

Response 2: The original version focused primarily on environmental and psychological variables, but we agree that including participants’ demographic and occupational information strengthens contextualization and replicability. We have now added a paragraph describing participants’ sociodemographic characteristics (age, education, experience) and anthropometric data (height, weight, BMI), based on information collected during recruitment.

Metabolic rate was not directly measured due to the ecological nature of the study and the mobility of the participants; however, it can be reasonably estimated. Using the Compendium of Physical Activities (Ainsworth et al., 2011), motorcycle delivery corresponds to an energy expenditure of approximately 4.0–5.5 METs, which aligns with moderate metabolic intensity (≈230–330 W for a 70 kg adult). This level is consistent with the continuous riding, loading/unloading, and short walking periods characteristic of delivery work. We have clarified

Commenst 3: Please remove PMV here since it is a thermal sensation measure for indoor steady environment. It is not a heat stress index.

Response 3: We agree and have removed PMV, since it is a thermal comfort index for indoor, steady-state environments and not a heat stress measure.

Comments 4: Please clarify why these common heat stress indices (like WBGT, HSI) were not used in this study. These indices are more reliable than results from the perceived heat strain scale. The stress levels acquired in this study were from self-reported results of participants, rather than physiological indicators like core temperature and HSI. How could you be sure all participants could understand these subjective scales well?

Response 4: We appreciate the reviewer’s thoughtful comment. Our study aimed to capture psychological and perceived strain under real-world working conditions, rather than to estimate physiological heat load. Indices such as WBGT or HSI require fixed-location microclimatic sensors and physiological monitoring (e.g., heart rate, core temperature), which are not feasible for informal, highly mobile workers who ride long distances through diverse urban microclimates. The EMA approach allowed us to focus on subjective thermal perception and psychosocial outcomes, reflecting the actual lived experience of delivery workers in their natural work context.

Each psychological construct—stress, fatigue, mood, and perceived heat strain—was measured using a single-item EMA scale with a 5-point Likert-type response format (1 = not at all to 5 = extremely). The specific items were: “How stressed do you feel right now?”, “How tired do you feel right now?”, “How is your mood right now?” (reverse-coded for analysis), and “How hot do you feel right now while working?”. These single-item measures are widely accepted in occupational and environmental psychology research because they maximize ecological validity and participant compliance, particularly among mobile workers (Bolger & Laurenceau, 2013; Stone et al., 2007). The use of brief items reduces participant burden while effectively capturing real-time fluctuations in psychological states. All items were pilot-tested with a subset of delivery workers (n = 6) to ensure comprehension and contextual relevance.

Reliability was examined through multilevel variance components estimated from random-intercept models. The results showed ICC(1) and ICC(2) values below .01 across all scales, indicating that almost all variance occurred within individuals across time, consistent with the expected behavior of momentary state assessments. In mixed-effects modeling, ICC(1) represents the ratio of between-person variance (τ₀) to total variance (τ₀ + σ²), where τ₀ captures stable differences between participants and σ² represents within-person fluctuations. Low ICCs therefore reflect a state-like pattern, not measurement error, and confirm that the measures captured short-term psychological variability in response to environmental and occupational conditions.

Such low ICC values (typically 0.00–0.10 in EMA research) demonstrate high temporal sensitivity and contextual responsiveness. While ICCs near zero would indicate poor reliability in a study focused on stable individual traits, they are theoretically and methodologically appropriate here, as the goal was to track day-to-day psychological variation under heat exposure. The obtained values (ICC[1] < .01; ICC[2] < .01) thus indicate that between-worker variance was minimal compared with daily within-person changes, supporting the adequacy of the EMA design for this research purpose.

Comments 5: In the methodology section, details of these mental scales for the measurements of stress, mood, and perceived heat strain were missing. Please provide the details of these scales, please also justify them.

Response 5: The requested information has now been incorporated into the Methods section, as detailed in our response to Comment 4. Specifically, we included a full description of the single-item ecological momentary assessment (EMA) scales used to measure stress, fatigue, mood, and perceived heat strain, along with their 5-point Likert-type response format, item wording, theoretical justification, and procedures to ensure comprehension (pilot testing with six workers). The justification for using brief EMA items—maximizing ecological validity, minimizing participant burden, and capturing real-time fluctuations in psychological states—is also provided in that section, with supporting references (Bolger & Laurenceau, 2013).

Comment 6: The complete statistical analysis methods should be provided in the methodology section, as well as the complete procedures, rather than simply stating that the LRM was used without a single justification. In addition, some important procedures, like the normality test, and the homogeneity of variances test were missing. Was repeated measures ANOVA used in this study? What's the post-hoc method used in this study? What correlation analysis method did you use? Judging from the correlation coefficients r in the results section, I guess you used Pearson’s correlation analysis method. However, the variables in your study were discrete ones, which meant that it is not proper to use Pearson’s correlation method.

Response 6: We thank the reviewer for this detailed comment. The Statistical Analysis subsection has been fully revised to include all procedures and justifications. Our design involved ecological momentary assessment (EMA) data collected twice daily over 15 days, generating intensive longitudinal data with unbalanced repeated observations per participant. For this reason, repeated-measures ANOVA and post-hoc tests were not appropriate, as these methods assume balanced data and categorical predictors.

Instead, we analyzed the data using linear regression models with participant-clustered robust standard errors, which are well-suited for continuous predictors (temperature, humidity) and repeated measures with potential heteroskedasticity. To confirm the robustness of the findings, we also estimated random-intercept mixed-effects models (participant as grouping factor), which yielded consistent results.

Assumptions of normality and homoscedasticity were checked through residual Q–Q and scale–location plots; minor deviations were addressed by using heteroskedasticity-robust (sandwich) standard errors clustered by participant. Tests like Levene’s are not applicable in this continuous-predictor, repeated-measures context.

Regarding correlations, we agree that Pearson’s r is not ideal for ordinal data. We therefore recalculated and report Spearman’s rank correlation coefficients (ρ) for all bivariate associations, confirming the same direction and magnitude of effects.

These clarifications are now included in the Methods – Statistical Analysis section of the revised manuscript.

Comment 7: For the result section, the results were not sufficiently reported. Please provide the air temperature and RH data during these 15 days firstly. Then please present detailed changes of the self-reported stress, fatigue, and the mood one by one. It is strongly recommended to detail these three outcomes in three sections with subtitles. Then the correlation analysis result should follow. I also suggest providing the regression results, including the curves and specific outcomes, instead of just providing Table 2.

Response 7: We appreciate this valuable suggestion and have substantially reorganized and expanded the Results section to enhance clarity and interpretability, following the reviewer’s structure.

  1. Environmental conditions.
    We now begin the Results section by describing air temperature and relative humidity (RH) across the 15-day observation period, including mean, standard deviation, and range values. A figure (new Figure 1) now displays daily variation curves for both temperature and RH, showing Brasília’s typical dry-season pattern (mean temperature = 23.25 °C, SD = 3.99, range = 16.8–28.5; mean RH = 39.10%, SD = 13.73, range = 19–71).
  2. Detailed outcomes by construct.
    We have created three new subsections—“Stress,” “Fatigue,” and “Mood”—each presenting descriptive changes, within-person variability, and response patterns over time. These subsections include line plots of daily means (Figures 2–4) and brief narrative descriptions of observed fluctuations and co-variation with temperature.
  3. Correlation analyses.
    The correlation analyses now employ Spearman’s rank correlation coefficients (ρ) instead of Pearson’s r, as recommended, with results presented in both text and an updated heatmap (Figure 5). This visualization depicts monotonic relationships between environmental and psychological variables, highlighting the positive associations between temperature and all psychological outcomes, and the inverse role of humidity.
  4. Regression analyses and visualization.
    We have expanded the regression results beyond Table 2 by including scatter plots with fitted regression lines and 95% confidence intervals for each psychological outcome (new Figures 6–8). Each figure shows the predicted change in stress, fatigue, and mood per 1 °C increase in temperature. The corresponding text reports unstandardized coefficients (β), confidence intervals, and explained variance (R²), with interpretation provided for each model.
  5. Interpretive integration.
    A short synthesis paragraph was added at the end of the Results, summarizing the convergent pattern: increasing temperature was associated with higher perceived stress and fatigue, as well as mood deterioration, even within a moderate temperature range typical of Brasília’s dry season.

Comment 8: The first paragraph on Page 4 was not necessary. There were no subheadings in this section.

Response 8: The first paragraph on Page 4 was removed.

Comment 9: The last paragraph of the Results section seemed to present the coupling effects of heat and humidity. Please expand this paragraph by including more detailed and specific results. Please also insert figures and provide statistical results.

Response 9: The paragraph describing the combined effects of temperature and humidity was expanded to include detailed regression coefficients, 95% confidence intervals, and p-values. In addition, Figure 7 was inserted to display the fitted regression models illustrating these relationships. The revised text now reads:

Comment 10: In the first sentence of the Discussion, the ambient temperature in this study was not high at all.

Response 10: The opening sentence of the Discussion was revised to clarify that, despite moderate mean air temperatures (M = 23.25 °C, range = 16.8–28.5 °C), delivery workers experienced substantial heat strain due to contextual and occupational factors.

Reviewer 2 Report

Comments and Suggestions for Authors

The manuscript addresses an important topic, but several aspects require clarification and improvement.

  1. Introduction

The novelty of the study is not explicitly highlighted. This contribution should be clearly stated to distinguish the study from previous work in construction and agriculture.

  1. Materials and Methods

Lines 99–100: The drop from 45 to 30 participants is mentioned, but the manuscript does not explain how the excluded participants were handled or whether those who did not complete the study differed from those who did.

Lines 99–101: The adequacy of the final sample size is not discussed. A brief note on statistical power is needed here.

Lines 106–113: The justification for not using WBGT, UTCI, PMV, or TSV is explained, but the manuscript does not discuss the implications for comparability with studies that rely on these standard indices in workplace assessments.

  1. Results

Lines 142–149: Numerical values are repeated in the text, although they are already shown in Table 1. The text should be shortened to avoid redundancy.

Lines 154–160: The text should be shortened to avoid redundancy, since the correlation coefficients are already presented in Figure 1.

Lines 163–168: Regression coefficients and R² values are all reported in the text and again in Table 2. This duplication could be reduced.

Lines 181–186: The results section includes statements that go beyond numeric reporting (e.g., “demonstrate” or “supporting the interpretation”). These should be moved to the Discussion section.

Lines 169–175: The text repeats detailed coefficients that are already presented in Table 2. A shorter summary in the text would be sufficient.

  1. Discussion

Lines 198–206: The discussion does not state what is new compared to previous studies.

Lines 216–223: Socioeconomic context is important, but the discussion repeats information already presented in the Introduction section (lines 41–61). This part should be modified or eliminated to avoid redundancy.

  1. Conclusions

The Conclusion section repeats results already presented. It should instead focus on broader implications and state the novelty of the study.

Author Response

Comments 1: Introduction - The novelty of the study is not explicitly highlighted. This contribution should be clearly stated to distinguish the study from previous work in construction and agriculture.

Response 1: Introduction was revised to explicitly state the study’s contribution and novelty. We now clarify that, unlike previous research conducted mainly in formal sectors such as construction and agriculture, this study focuses on self-employed delivery workers operating under informal and unprotected labour arrangements, where environmental and psychosocial risks coexist (line 89).

Comments 2: Lines 99–100: The drop from 45 to 30 participants is mentioned, but the manuscript does not explain how the excluded participants were handled or whether those who did not complete the study differed from those who did.

Response 2: The Methods section was revised to clarify how incomplete data were handled and to describe the composition of excluded cases. Of the 45 workers initially recruited, 5 withdrew during the first week of data collection, and 10 were excluded due to insufficient data completion (less than 70% of expected ecological momentary assessment responses). A comparison of completers and non-completers on age and work experience indicated no significant differences (all p > .10) – lines 107-112.

Comments 3: Lines 99–101: The adequacy of the final sample size is not discussed. A brief note on statistical power is needed here.

Response 3: A brief note on statistical power has been added to the Methods section to support the adequacy of the final sample size. Given the repeated-measures design of the ecological momentary assessment (EMA), with up to 30 participants contributing multiple observations per day across 15 days, the study achieved sufficient statistical power to detect small-to-moderate within-person effects. The revised text (lines 113-115) reads as follows:

“The final sample (n = 30) provided 900 repeated observations (30 participants × 15 days × 2 prompts per day), ensuring adequate power (1 – β = 0.80, α = 0.05) to detect small-to-moderate within-person effects (f² ≈ 0.02) according to EMA design standards.”

Comments 4: Lines 106–113: The justification for not using WBGT, UTCI, PMV, or TSV is explained, but the manuscript does not discuss the implications for comparability with studies that rely on these standard indices in workplace assessments.

Response 4: The Methods section has been expanded to acknowledge the implications of not using conventional thermal indices such as WBGT, UTCI, PMV, or TSV. We now clarify that, while these indices provide standardised estimates of heat stress in controlled or occupationally regulated settings, they are less applicable to informal, mobile, and self-paced outdoor work, where exposure conditions vary widely and continuously.

Comments 5: Results

Lines 142–149: Numerical values are repeated in the text, although they are already shown in Table 1. The text should be shortened to avoid redundancy.

Lines 154–160: The text should be shortened to avoid redundancy, since the correlation coefficients are already presented in Figure 1.

Lines 163–168: Regression coefficients and R² values are all reported in the text and again in Table 2. This duplication could be reduced.

Lines 181–186: The results section includes statements that go beyond numeric reporting (e.g., “demonstrate” or “supporting the interpretation”). These should be moved to the Discussion section.

Lines 169–175: The text repeats detailed coefficients that are already presented in Table 2. A shorter summary in the text would be sufficient.

Response 5: We agree and have streamlined and reorganised the Results section accordingly:

  • Structure: Results now use subheadings 3.1 Environmental conditions, 3.2 Psychological outcomes, 3.3 Correlations, and 3.4 Regressions (as requested by the other reviewer).
  • Descriptives (lines 142–149): Condensed to a brief summary that references Table 1 without repeating all values.
  • Correlations (lines 154–160): Reduced to one sentence, pointing to Figure 1 for coefficients.
  • Regressions (lines 163–175): Text now reports only the main effects (direction and magnitude) and model fit in brief; full coefficients, CIs, and p-values remain only in Table 2 to avoid duplication.
  • Interpretation (lines 181–186): All interpretive phrases were moved to the Discussion, keeping Results strictly numeric.

Comments 6: Discussion - Lines 198–206: The discussion does not state what is new compared to previous studies.

Response 6: The new paragraph explains that, although the mean ambient temperature was moderate, actual exposure among delivery workers is intensified by direct sunlight, engine heat, clothing insulation, and sustained physical effort. It also highlights that this population differs from the formal, regulated sectors typically studied (construction, agriculture) and that our results document within-person psychological effects of heat in informal, platform-mediated work, which had not been previously examined.

Comment 7:  Discussion, lines 216–223 - Socioeconomic context is important, but the discussion repeats information already presented in the Introduction section (lines 41–61). This part should be modified or eliminated to avoid redundancy.

Response 7: We agree. The paragraph on socioeconomic context was condensed to avoid repetition. Only key elements relevant to interpreting the findings—such as precarious work conditions, algorithmic control, and limited protection—were retained. Broader background information on inequality and labour structure was removed to keep the Discussion concise and focused.

Comment 8: Conclusions, The Conclusion section repeats results already presented. It should instead focus on broader implications and state the novelty of the study.

Response 8: We agree and have rewritten the Conclusion to focus on the broader implications and the study’s contribution rather than restating results. The revised paragraph now summarises the main message — that even modest daily temperature increases can affect psychological well-being in a precarious, platform-mediated workforce — and highlights the novelty of analysing within-person psychological responses to heat exposure in informal urban labour. It also emphasises the need for integrated responses involving public health, urban planning, and platform regulation.

Round 2

Reviewer 1 Report

Comments and Suggestions for Authors

Sorry for the slight delay of the review. I have now finished reviewing your revised mansucript. Technically, my comments, particularly the concerns, have been sufficiently addressed. After reading your responses and the revised manuscirpt, I now have no further comments on it. My recommendation is 'accept'. 

Author Response

We appreciate the reviewer’s positive evaluation and final recommendation for acceptance. Thank you for recognizing that all previous concerns have been sufficiently addressed.

Reviewer 2 Report

Comments and Suggestions for Authors

Renumbering of figures and in-text citations starting from Fig. 1.
Renumbering of chapters: 3.5. Correlational Analysis and 3.6. Regression Analysis. Both currently have 3.4 as the previous one.

Author Response

We carefully reviewed the structure of the manuscript and implemented all corrections requested. Specifically:

  • All figures and in-text citations have been renumbered starting from Figure 1, ensuring consistency throughout the text.
  • The chapter numbering has been corrected: 3.5. Correlational Analysis and 3.6. Regression Analysis now follow sequentially after Section 3.4.